# Single-Shot Plug-and-Play Methods for Inverse Problems

**Yanqi Cheng**[1], **Lipei Zhang**[1*], **Zhenda Shen**[2*], **Shujun Wang**[3], **Lequan Yu**[4], **Raymond H. Chan**[5], **Carola-Bibiane Schönlieb**[1], **Angelica I Aviles-Rivero**[6,1]

[1]*Department of Applied Mathematics and Theoretical Physics, University of Cambridge*
[2]*Department of Mathematics, City University of Hong Kong*
[3]*Biomedical Engineering, Hong Kong Polytechnic University*
[4]*Department of Statistics and Actuarial Science, The University of Hong Kong*
[5]*School of Data Science, Lingnan University*
[6]*Yau Mathematical Sciences Center, Tsinghua University*

**Reviewed on OpenReview:** https://openreview.net/forum?id=vXevE43NxF

## Abstract

The utilisation of Plug-and-Play (PnP) priors in inverse problems has become increasingly prominent in recent years. This preference is based on the mathematical equivalence between the general proximal operator and the regularised denoiser, facilitating the adaptation of various off-the-shelf denoiser priors to a wide range of inverse problems. However, existing PnP models predominantly rely on pre-trained denoisers using large datasets. In this work, we introduce Single-Shot PnP methods (SS-PnP), shifting the focus to solving inverse problems with minimal data. First, we integrate Single-Shot proximal denoisers into iterative methods, enabling training with single instances. Second, we propose implicit neural priors based on a novel function that preserves relevant frequencies to capture fine details while avoiding the issue of vanishing gradients. We demonstrate, through extensive numerical and visual experiments, that our method leads to better approximations.

## 1 Introduction

Inverse problems have long been a fundamental challenge in the field of mathematics and applied sciences, encompassing a wide range of applications from image reconstruction to signal processing (Devaney, 2012; Bertero et al., 2021). Traditionally, these problems have been approached through various analytical and numerical methods (Vogel, 2002; Jordan, 1881; Metropolis & Ulam, 1949) using either single or multiple images (without a deep net or learning process). However, the advent of deep learning has revolutionised this domain. Deep inverse problems present a modern approach, offering new insights and solutions where conventional methods have limitations (McCann et al., 2017). This shift towards leveraging machine learning techniques marks a significant evolution in tackling inverse problems, opening doors to more sophisticated and efficient problem-solving techniques.

A popular framework in this deep inverse problem era is Plug-and-Play (PnP) methods (Venkatakrishnan et al., 2013; Zhang et al., 2021; Chan et al., 2016; Ono, 2017). At the core of this approach lies the mathematical equivalence of the proximal operator to denoising (Venkatakrishnan et al., 2013), a concept that intertwines optimisation theory with modern denoisers. This equivalence paves the way for the integration of advanced deep learning-based denoisers into the inverse problem-solving process. The Plug-and-Play framework essentially allows for the seamless insertion of these denoisers into iterative algorithms, enhancing their ability to recover high-quality signals or images from corrupted observations (Zhang et al., 2021; 2019; Ahmad et al., 2020).

---

*Joint contribution.

Although PnP methods have demonstrated outstanding results across a wide range of inverse problems, they predominantly rely on the assumption of having substantial training datasets for the development of robust denoising models (Arridge et al., 2019). This requirement often becomes a significant bottleneck, especially in scenarios where data availability is limited or the diversity of data is not sufficiently representative. Moreover, PnP methods typically necessitate retraining or fine-tuning to adapt effectively to varying distributions and signal characteristics. To address these challenges, Single-Shot learning emerges as a promising alternative, offering a paradigm shift in how deep inverse problems can be trained with minimal data. Unlike traditional methods that require extensive datasets, Single-Shot learning aims to make significant inferences from a single instance, or in some cases, a small set of instances. *To the best of our knowledge, there is no existing work on Single-Shot PnP methods. Our work thus opens the door to a novel research line for PnP, introducing the concept of Single-Shot Plug-and-Play methods.*

Single-Shot techniques offer a viable solution to the current constraints in PnP methods, particularly in their reliance on extensive training datasets. This shift enables us to delve into the recent advancements in signal representation, specifically, the emergence of neural implicit representations. Neural implicit representations (Strümpler et al., 2022; Saragadam et al., 2023; Tancik et al., 2020) are trained to learn continuous functions that map coordinates to signal values or features, making this form of representation highly efficient in capturing intricate data details with a compact network architecture. Its application in the context of Single-Shot PnP methods is particularly promising, as denoisers that can effectively operate with limited data inputs, aligning with the Single-Shot learning paradigm. *We therefore use implicit neural representations as a way to represent the proximal denoiser in PnP techniques in a Single-Shot fashion.* Our contributions are:

✧ We propose the concept of Single-Shot Plug-and-Play (SS-PnP) Methods to solve inverse problems, in which we highlight:

- We introduce Single-Shot proximal denoisers into iterative methods for solving any inverse problem. Our scheme eliminates the need for a pre-trained model, enabling training with a single instance.
- We introduce implicit neural priors for Plug-and-Play methods that enables the network to preserve more details during training. Additionally, we provide a theoretical justification for our prior, emphasising how its continuity and differentiability play a crucial role in mitigating the issue of vanishing gradients during the training process and preserve fine details.

✧ We demonstrate, through extensive experiments on several inverse problems, that our technique leads to a better approximation on capturing finer details, smoother edge features and better colour representation. The method is evaluated on inverse problems with both single operator task and multi-operator task like joint demosaicing and deconvolution task. With only one image input in the whole reconstruction process, it outperforms the classical methods and pre-trained models among all the tasks.

## 2 Related Work

In this section, we review the existing literature and the concepts closely related to our work.

**Plug-and-Play (PnP) Methods.** They have revolutionised the field of inverse problems by integrating advanced denoisers into iterative algorithms. This innovative approach, initiated by Venkatakrishnan et al. (2013), has undergone significant evolution. Meinhardt et al. (2017) showcased its effectiveness in diverse imaging tasks, marking a turning point in PnP's development. The subsequent works (Ryu et al., 2019; Sun et al., 2019; Hurault et al., 2022) further refined PnP, enhancing its stability and convergence, thus broadening its applicability.

In addition, the works of that (Teodoro et al., 2018; Yuan et al., 2020; Zhang et al., 2017b; Ono, 2017; Sun et al., 2019) further expanded the scope of PnP, demonstrating its adaptability in complex imaging scenarios. A notable advancement in the optimisation landscape came with the introduction of TFPnP (Tuning-Free

Plug-and-Play) (Wei et al., 2020), which innovatively eliminated the need for parameter tuning in PnP algorithms.

Previous methods have relied on data-driven pre-training, which becomes impractical in situations with limited data or on smaller devices due to the extensive size of the required models. Consequently, developing a method for Single-Shot image prior denoising emerges as a compelling solution for such resource-constrained environments.

A wide range of denoisers has been used within the PnP framework. Classical denoisers such as BM3D (Dabov et al., 2007) have been the most prevalent. Other notable approaches include Teodoro et al. (2016) and Venkatakrishnan et al. (2013). These traditional denoisers are well-established and often require little or no data pre-training. On the other hand, another emerging family of denoisers leverages deep learning techniques (Meinhardt et al., 2017; Zhang et al., 2017b; Laumont et al., 2022). These deep learning-based denoisers have gained prominence for their ability to capture complex features and patterns in data, making them highly effective in PnP frameworks. The aim of this paper is to further explore and advance the use of deep learning-based denoisers, particularly in scenarios with minimal data, through the proposed Single-Shot methodology.

**Single-Shot Image Denoising.** A crucial element in PnP methods is the denoiser model. During the last years, the denoiser technique in PnP has evolved remarkably, transitioning from traditional methods to advanced deep learning techniques. Pioneering works such as the BM3D algorithm (Dabov et al., 2007) and the Non-Local Means (NLM) algorithm (Buades et al., 2005) laid the groundwork, setting significant benchmarks. The introduction of deep learning marked a paradigm shift, exemplified by the DnCNN model proposed by Zhang et al. (2017a) and its variant DnCNN-S (Zhang et al., 2018), which demonstrated the efficacy of convolutional networks in denoising. The Deep Image Prior (DIP) by Lempitsky et al. (2018) furthered this progression, utilising the structure of convolutional networks as a prior for denoising.

A notable advancement is the rising of self-supervised methods, which revolutionised the Single-Shot denoising field by eliminating the need for clean training data. The Noise2Void framework by Krull et al. (2019) and the Noise2Self algorithm (Batson & Royer, 2019) are pioneering examples, utilising concepts like blind-spot networks for effective denoising. Building on these, Self2Self (Quan et al., 2020), Noise2Same (Xie et al., 2020), and Noise2Info (Wang et al., 2023) further explored self-supervision, offering unique strategies for leveraging the inherent properties of noisy images. Additional approaches like CycleISP (Zamir et al., 2020), IRCNN (Zhang et al., 2017b), GCDN (Anwar & Barnes, 2020), and bayesian denoising with blind-spot networks (Laine et al., 2019) further enrich the landscape, each contributing novel perspectives and solutions to the challenge of Single-Shot image denoising.

These advancements are not confined to noise reduction alone; many of the developed deep denoisers are inherently adaptable and can be generalised to tackle various inverse problems in imaging. Inverse problems, such as demosaicing, denoising, and deconvolution, share common traits with denoising. The underlying principles and network architectures developed for denoising can often be extended or fine-tuned to address these challenges (Romano et al., 2017; Akyüz et al., 2020). Furthermore, the concept of Plug-and-Play methods opens up new avenues. *However, this progress highlights a gap: despite the evolution of Single-Shot denoisers, there is currently no work on Single-Shot denoisers into iterative algorithms. Therefore, this work introduces the concept of Single-Shot Plug-and-Play methods.*

Despite the versatility of Single-Shot image denoising, CNN-based estimators frequently fail to capture continuously high-frequency details crucial for image reconstruction. Implicit neural representation (INR) emerges as a solution, adept at addressing these high-frequency challenges in inverse problems.

**Implicit Neural Representation**, characterised as a fully connected network-based method, has seen a rise in popularity for solving inverse problems as highlighted by Sun et al. (2021a). Traditional activation functions like ReLU have exhibited limitations in representing high-frequency features, as discussed by Dabov et al. (2007). This shortcoming has led to the exploration of nonlinear activation functions, such as the sinusoidal function (Sitzmann et al., 2020), enhancing representational capabilities. The adaptability of INR is evident in its diverse applications across medical imaging (Wang et al., 2022), image processing (Chen et al., 2021; Attal et al., 2021), and super-resolution (Saragadam et al., 2023).

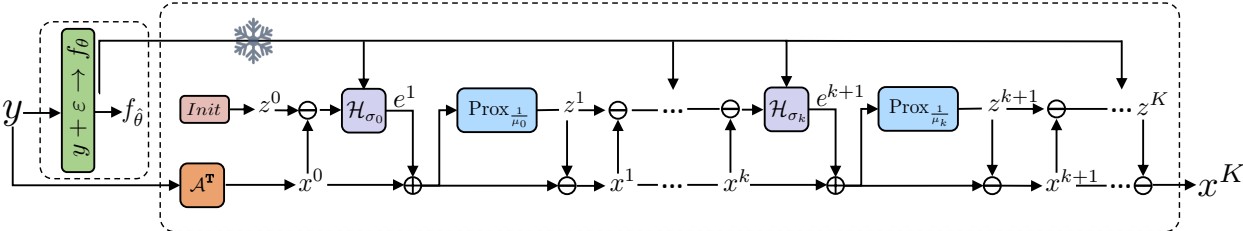

Figure 1: The pipeline of Single-Shot Plug-and-Play methods (SS-PnP). The 2 blocks indicate the 2 steps as in Algorithm 1, and the ❄ indicates the fix in denoiser weight over the ADMM iterations, where there are $K$ iterations in total (i.e. $k \in \{0, 1, ..., K - 1\}$).

A distinct advantage of INR lies in its independence from pre-training, attributed to its rapid training capability, as indicated by Saragadam et al. (2023). This feature makes INR particularly suitable for Single-Shot image denoising tasks, where a single image suffices for effective learning, bypassing the need for extensive data-driven pre-training. The ability of INR to learn from minimal data points underscores its potential in resource-limited scenarios.

## 3 Methodology

Inverse imaging problems have emerged as a crucial cornerstone in computer vision and various related domains. A conventional forward model reads:

$$y = A(x) + \epsilon, \tag{1}$$

where $y \in \mathbb{R}^M$ is the known observation, $x \in \mathbb{R}^N$ is the unknown target of interest, $\epsilon \in \mathbb{R}^M$ is the measurement noise, and $A : \mathbb{R}^N \to \mathbb{R}^M$ is the forward measurement operator, which varies across different tasks.

The approximation of $x$ gives rise to a highly ill-posed problem, necessitating regularisation. Consequently, the fundamental objective for any imaging inverse problem is the minimisation of:

$$\min_{x \in \mathbb{R}^N} D(x) + \gamma R(x), \tag{2}$$

$D(x)$ refers to the data fidelity term, usually taking the form $D(x) = \frac{1}{2} \|A(x) - y\|^2$. $R(x)$ denotes the regularisation term, which encodes prior knowledge about $x$. The parameter $\gamma$ serves as the regularisation weight, determining the trade-off between data fidelity and regularisation. A widely used approach to solve equation 2 is the family of first-order optimisation methods (Beck & Teboulle, 2009b; Boyd et al., 2011; Chambolle & Pock, 2011).

### 3.1 Single-Shot Proximal Denoiser

We introduce Single-Shot Plug-and-Play methods (SS-PnP) as demonstrated in Figure 1. The optimisation of equation 2 typically exhibits non-smooth characteristics due to $R$. A widely adopted strategy for addressing this problem is to employ first-order methods such as the alternating direction method of multipliers (ADMM). Given a function $F(\cdot)$, we define the proximal operator of $F$ at $v$ with step size $\delta$ as:

$$\text{Prox}_{\delta F}(v) = \underset{u}{\text{argmin}} \left\{ \frac{1}{2} \|u - v\|^2 + \delta F(u) \right\}, \tag{3}$$

Considering ADMM, one can express PnP-ADMM as:

$$\begin{aligned}
\bigstar \ e^{k+1} &= \text{Prox}_{\sigma_k^2 R} \left( z^k - x^k \right) = \mathcal{H}_{\sigma_k}(z^k - x^k) \\
z^{k+1} &= \text{Prox}_{\frac{1}{\mu_k} D} \left( e^{k+1} + x^k \right) \\
x^{k+1} &= x^k + e^{k+1} - z^{k+1}
\end{aligned} \tag{4}$$

---

**Algorithm 1:** Single-Shot Plug-and-Play

---

**Forward model:** $y = A(x) + \epsilon$

**Input:**   $y \in \mathbb{R}^M$

**Output:**  $x \in \mathbb{R}^N$

---

**Step 1**  Train the Single-Shot Denoiser

Choose noise strength $\varepsilon$, and initialise $f_{\theta_0}$                           ▸ $\varepsilon$ is independent of $\epsilon$

**for** $i \in \{0, 1, ..., I-1\}$ **do**

$\quad\Big|\quad f_{\theta_{i+1}} \leftarrow \mathcal{L}(f_{\theta_i}, y + \varepsilon)$                        ▸ Train the Single-Shot denoising prior

**end**

$f_{\hat\theta} = f_{\theta_I}$

**Step 2**  ADMM Iteration

Let $\mathcal{H} = f_{\hat\theta}$ , and $x^0 = A^T(y)$                              ▸ $A^T$ denotes adjoint of $A$

Choose noise strength $\sigma : \{\sigma_k\}_0^{K-1}$ , and penalty parameter $\mu : \{\mu_k\}_0^{K-1}$

Initialise $z^0$

**for** $k \in \{0, 1, 2, ..., K-1\}$ **do**

$\quad\Big|\quad e^{k+1} = \mathcal{H}_{\sigma_k}(z^k - x^k)$

$\quad\Big|\quad z^{k+1} = \text{Prox}_{\frac{1}{\mu_k}D}\left(e^{k+1} + x^k\right)$

$\quad\Big|\quad x^{k+1} = x^k + e^{k+1} - z^{k+1}$

**end**

$x = x^K$

---

where $\text{Prox}_{\sigma_k^2 R}(\cdot)$ is the proximal operator of the regularisation with noise strength $\sigma_k$ and $\text{Prox}_{\frac{1}{\mu_k}D}(\cdot)$ is to enforce the data consistency (Ryu et al., 2019) with penalty parameter $\mu$ in the $k$-th iteration, for $k \in \{0, 1, 2, ..., K-1\}$. From equation 4 -★, we can observe that Plug-and-Play (PnP) methods leverage the equivalence between the proximal operator $\text{Prox}_{\sigma_k^2 R}(\cdot)$ and a denoiser $\mathcal{H}_{\sigma_k}$ with the denoising parameter $\sigma_k \geq 0$.

**Single-Shot Denoiser is All You Need for PnP.** PnP techniques primarily rely on denoisers, often trained on extensive datasets and using off-the-shelf deep denoisers (Ryu et al., 2019; Sun et al., 2019; Hurault et al., 2022). However, an unexplored question is *whether Single-Shot denoisers can be used in iterative methods and what their properties are. To our knowledge, there are no existing works on this.*

In our Single-Shot denoising stage, we utilise the observed image with noise $y + \varepsilon$ to train the denoiser, enabling it to distinguish and mitigate complex noise and distortions specific to the example. The neural network $f_\theta$ aims to transform the noisy and corrupted image into its less corrupted counterpart. This is achieved through an optimisation process given by:

$$\hat\theta = \underset{\theta}{\text{argmin}}\, \mathcal{L}(f_\theta, y + \varepsilon), \tag{5}$$

where $\mathcal{L}$ is a loss function that evaluates the difference between the network's output and the corresponding observed image with noise. Refer to Step 1 in Algorithm 1.

This pre-trained model is subsequently integrated into a Plug-and-Play (PnP) framework as a prior for regularisation. Let $\mathcal{H} = f_{\hat\theta}$, the trained denoiser serves as a guiding force in the iterative reconstruction process, enhancing the ability to recover high-quality images from corrupted observations. By embedding this Single-Shot learning model into the PnP framework, we establish a potent approach for tackling a range

of inverse problems, particularly in situations involving severely corrupted images. To promote the synergy between Single-Shot learning and PnP methods, the $f_\theta$ utilise nonlinear INR denoiser. Refer to Step 2 in Algorithm 1.

In this work, we consider ADMM justified by its well-known outstanding performance (Venkatakrishnan et al., 2013; Wei et al., 2022) in comparison to existing methods, its well-established convergence properties (Ryu et al., 2019), and its modular approach, which collectively ensure its effectiveness across various inverse problem scenarios.

### 3.2 Implicit Neural Prior for Plug-and-Play

Implicit neural representations enable the learning of continuous functions from a signal. State-of-the-art performance relies on deep denoisers like UNet (Ronneberger et al., 2015) and FFDNet (Zhang et al., 2018). While convolutional-based denoisers have shown impressive results, there is currently no research exploring the use of implicit neural representation (INR) as a prior for PnP methods. *INR can be used as a Single-Shot approach in equation 4, we then open the door to a new research direction for PnP.*

INR boasts remarkable expressive power and inductive bias, paving the way for innovative denoising approaches. Another significant contribution of this work is the introduction of a new Single-Shot framework for PnP. In particular, *we present a novel INR prior.* Unlike the majority of existing works, we provide a solid theoretical justification for its properties and behavior.

**Zooming into our Prior.** We next define our proposed implicit neural representation prior.

---

**Our INR Prior**

We define a nonlinear activation function and form our implicit neural representation, which reads:

$$\Phi(x) = \exp\{-(a_1 x + b_1)^2\} \sin(a_2 x + b_2) \tag{6}$$
$$+ \frac{1}{\exp\{-(a_1 x + b_1)\} + 1}$$

Our proposed activation function is nonlinear. Moreover, it has key properties of differentiability and continuity.

---

*Proof.* Let $g(x) = \exp\{-(a_1 x + b_1)^2\}$ and $f(x) = \sin(a_2 x + b_2)$. For $\forall c \in \mathbb{R}$,

$$\frac{(fg)(x) - (fg)(c)}{x - c}$$
$$= \frac{f(x)g(x) - f(c)g(x) + f(c)g(x) - f(c)g(c)}{x - c}$$
$$= \frac{f(x) - f(c)}{x - c}g(x) + f(c)\frac{g(x) - g(c)}{x - c} \tag{7}$$

Consequently,

$$\lim_{x \to c} \frac{(fg)(x) - (fg)(c)}{x - c}$$
$$= \lim_{x \to c}\left[\frac{f(x) - f(c)}{x - c}g(x) + f(c)\frac{g(x) - g(c)}{x - c}\right]$$
$$= \lim_{x \to c}\frac{f(x) - f(c)}{x - c}\lim_{x \to c}g(x) + f(c)\lim_{x \to c}\frac{g(x) - g(c)}{x - c} \tag{8}$$

Because both $g(x)$ and $f(x)$ are continuous and differentiable, $\lim_{x \to c} \frac{(fg)(x)-(fg)(c)}{x-c}$ exists for all $c \in \mathbb{R}$. Hence, $\exp\{-(a_1 x + b_1)^2\} \sin(a_2 x + b_2)$ is continuous and differentiable for all $c \in \mathbb{R}$. Consequently, the entire activation function is the sum of two differentiable functions, making it both continuous and differentiable. □

*Why are these properties interesting?* The nonlinearity, continuity, and differentiability brought by our formula make it more representative compared to traditional activation functions, enabling the network to preserve more details during training. Additionally, the continuity and differentiability of our formula play a crucial role in mitigating the issue of vanishing gradients during training.

---

**Convergence of our Prior**

Consider the function $\Phi(x)$ in equation 6, then it holds that:

$$\Phi(x) = \begin{cases} 0 & x \to -\infty \\ 1 & x \to \infty \end{cases} \tag{9}$$

This behavior ensures that $\Phi(x)$ remains bounded and avoids gradient explosion, making it a stable function for optimisation.

---

*Proof.* For $y > 0, \forall \epsilon > 0, \quad \exists \bar{y} = \sqrt{\max\left(0, \ln\frac{1}{\epsilon}\right)}$ when $y > \bar{y}$ ,

$$\begin{aligned}
\exp\left(y^2\right) &\sin(a_2 x + b_2) < \exp\left(-y^2\right) \\
&= \exp\left\{-\max\left(0, \ln\frac{1}{\epsilon}\right)\right\} \\
&= \exp\{\min(0, \ln\epsilon)\} \\
&\leqslant \exp(\ln\epsilon) = \epsilon
\end{aligned} \tag{10}$$

Meanwhile, the function $\frac{1}{\exp(-y)+1}$ converges to 1 when $x \to \infty$. Consequently, the function is convergent to 1 when $x \to \infty$. $\qquad\square$

We underline that previous proof that the convergence results we present are specifically in terms of the behaviour of the prior within our proposed framework. By leveraging the convergence and bounded nature of the activation function, we can significantly mitigate the issue of exploding gradients. Moreover, as the model converges to 0, it also effectively reduces the impact of outliers, enhancing the overall robustness of our model.

## 4 Experiment

In this section, we describe the experiments undertaken to validate our proposed Single-Shot Plug-and-Play framework.

### 4.1 Experimental Setting

In our image preprocessing, we utilised dual resizing strategies. We sourced the images with Creative Commons Licenses and resized to $512 \times 384$, and in the meantime tested on the selected data in Bevilacqua et al. (2012) and Zeyde et al. (2012), without resizing. We remind to the reader, the experiments on Single-Shot Plug-and-Play methods (SS-PnP) considered only a single image input in the whole pipeline.

**Training Scheme.** During the initial implicit neural representation (INR) pre-training phase, Gaussian noise with a standard deviation in the range of $[0.001, 0.5]$ was explored with 0.1 was set for all the experiments. The training was conducted over 100 iterations, with a network configuration comprising 2 hidden layers and 64 features per layer. The learning rate was set to 0.001. We then reconstruct the image for 5 ADMM iteration steps with dynamic noise strength and penalty parameter chosen by logarithmic descent that gradually decreases value between 35 and 30 over steps.

**Evaluation Protocol.** For comparative analysis, the Noise2Self pre-training scheme (Krull et al., 2019), was applied to three networks, each undergoes 100 training iterations with a learning rate of 0.01. The DnCNN (Zhang et al., 2017a) and FFDNet (Zhang et al., 2018) architectures, were configured with 8 hidden

Table 1: The performance (PSNR(dB)) comparison of the 4 Single-Shot deep denoising priors (DnCNN, FFDNet, UNet and our proposed ★) on super resolution (SR) task with 2× and 4× upscalings in the Plug-and-Play framework.

| SS-PnP | Fractal | | Wolf | | Dog | | Peacock | | Tiger | | Bird | |
|---|---|---|---|---|---|---|---|---|---|---|---|---|
| (Backbone) | 2× | 4× | 2× | 4× | 2× | 4× | 2× | 4× | 2× | 4× | 2× | 4× |
| DnCNN | 19.29 | 17.45 | 20.71 | 17.16 | 18.29 | 14.31 | 19.03 | 16.79 | 19.22 | 14.28 | 21.90 | 18.68 |
| FFDNet | 22.04 | 19.59 | 21.12 | 22.85 | 24.02 | 21.98 | 22.31 | 18.99 | 23.02 | 19.11 | 26.34 | 21.86 |
| UNet | 18.58 | 14.18 | 21.78 | 15.83 | 18.71 | 14.34 | 19.56 | 16.25 | 18.87 | 15.52 | 21.82 | 16.53 |
| ★ | **25.43** | **21.26** | **27.21** | **23.30** | **25.80** | **22.00** | **22.46** | **19.27** | **23.12** | **19.45** | **26.44** | **22.06** |

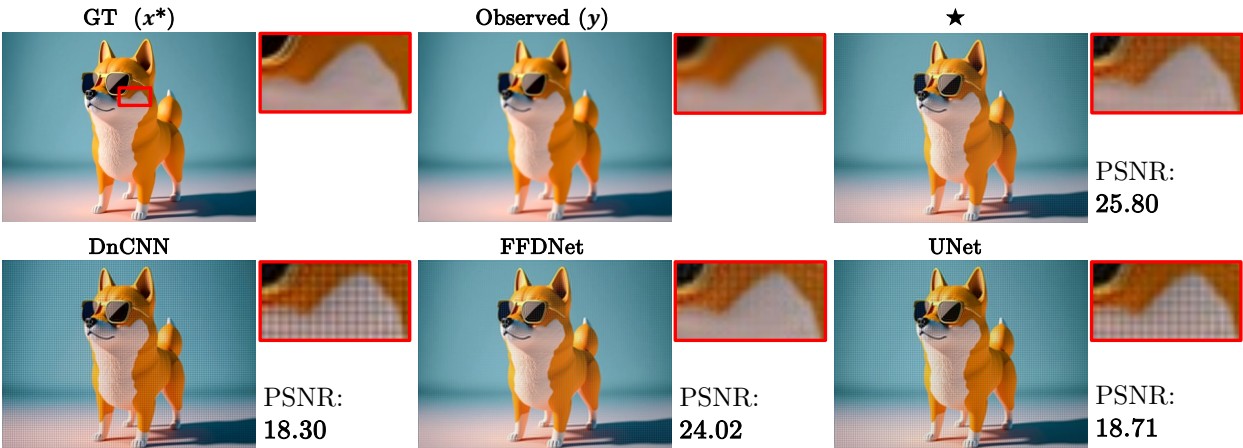

Figure 2: The comparative visualisation of super resolution with 2× upscaling task on "Dog" example among the 4 Single-Shot deep denoising priors (with ★ denotes our proposed prior) in Plug-and-Play framework. The zommed-in view is provided at the right hand side of each result.

layers and 64 feature maps per layer. Conversely, the UNet (Ronneberger et al., 2015) architecture, employed a 4-times downsampling strategy and initiated with 32 feature maps in first convolutional operation. For a fair comparison, we fixed the setting for the optimisation step as used in our proposed strategy.

The empirical studies are trained and tested on NVIDIA A10 GPU with 24GB RAM. The Plug-and-Play phase ensued with the ▽-Prox toolbox (Lai et al., 2023), adopting the default settings for all tasks. For evaluating our methods, we employed Peak Signal-to-Noise Ratio (PSNR) and Structural Similarity Index (SSIM). Higher PSNR and SSIM scores signal superior reconstruction quality.

### 4.2 Results & Discussion

This section shows all the numerical and visual results that support our method.

↻ **Super-resolution (SR).** It refers to the process of reconstructing a high-resolution image (HR) from one or more low-resolution observations (LR). The forward measurement operator for super-resolution is given by: $A(x) = (k \otimes x) \downarrow_s$, where $k$ represents the kernel for convoluting the image, and $\downarrow_s$ is the downscaling operation with scale $s$. Here, kernel operator was set as size 5 with standard deviation of the Gaussian distribution as 3.

In evaluating the Single-Shot super-resolution efficacy of our method, we utilise six varied categories of images, with upscaling factors of 2× and 4× to ensure a fair comparison against well-established techniques such as

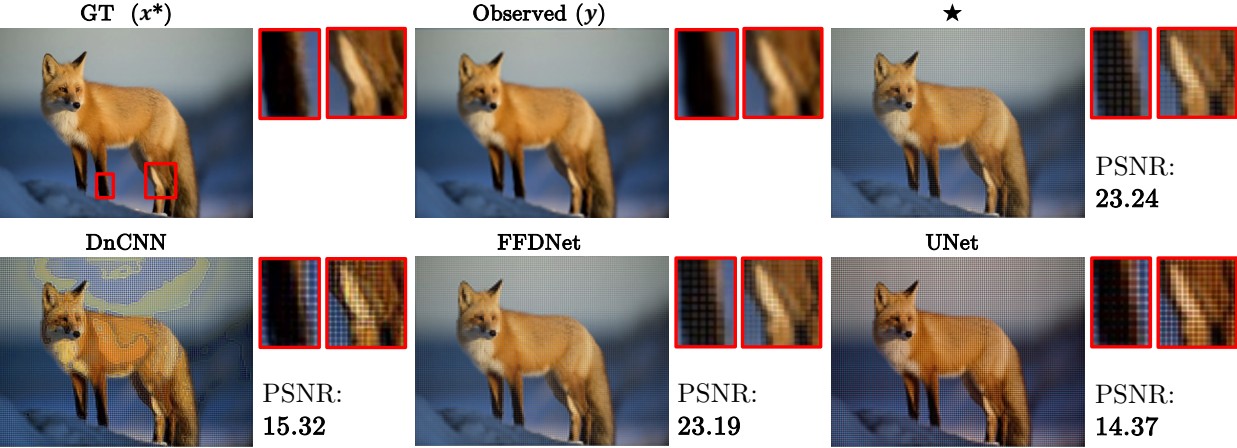

Figure 3: The visualisation comparison of super resolution with 4× upscaling task on "Fox" example between the 4 Single-Shot deep denoising priors performing within Plug-and-Play framework, with detailed comparison zoomed in. ★ denotes our prior.

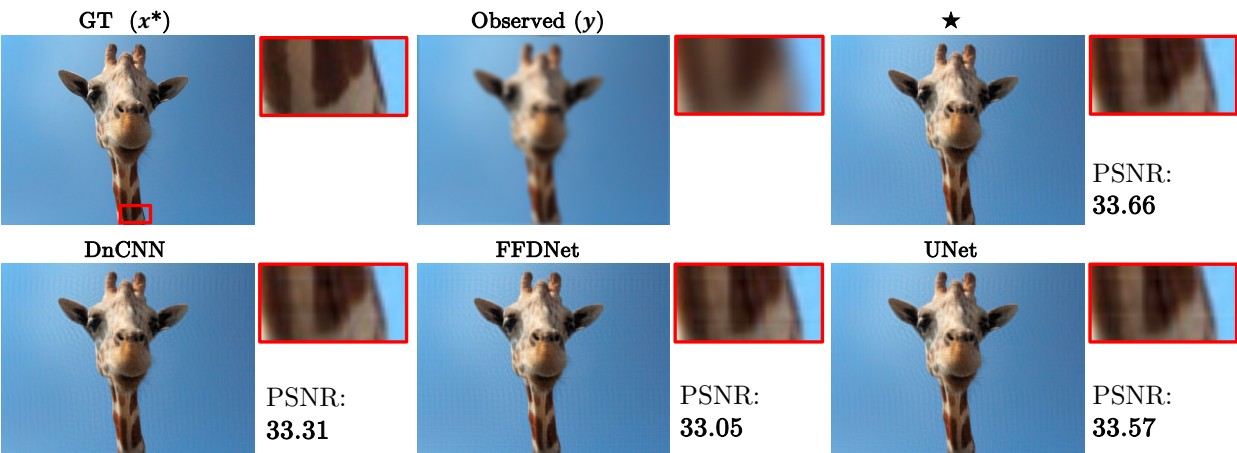

Figure 4: Visual comparison of the 4 deep denoising priors (include our proposed as ★) on deconvolution task in Single-Shot Plug-and-Play (SS-PnP) strategy for "Giraffe" example, with a zoomed-in region exhibiting intricate details.

DnCNN, FFDNet, and UNet, which are all trained with the Noise2Self pre-training scheme. The quantitative results, as detailed in Table 1, reveal our method's outstanding performance, outshining the benchmarks with significant margins. Notably, our approach demonstrates a marked improvement in PSNR values, with particularly pronounced enhancements in the 4x upscaling scenario. These advancements indicate our model's robustness and substantial improvement forward in Single-Shot super resolution.

Our method demonstrates exceptional preservation of texture and detail complexity, as shown in Figures 2, 3. Our technique achieves high-resolution enhancement while intricately reconstructing fine details, evident from the minimised artifacts such as grid patterns and spurious spots in Figure 2. In Figure 3, DnCNN introduces noticeable colour distortions in both the background and the foreground. The UNet generates numerous undesirable grids. Although our approach presents a comparable visual quality to FFDNet, it achieves a superior PSNR, suggesting a quantitatively and qualitatively improved performance. Collectively, our results indicate that our method not only holds promise for practical application but also sets a new standard for super-resolution tasks.

Table 2: Evaluation comparison of the 4 Single-Shot deep denoising priors within the Plug-and-Play framework (SS-PnP setting) on deconvolution task measured with PSNR(dB), and SSIM. We denote our proposed Single-Shot denoising prior as ★.

| SS-PnP | Racoon | | Turtle | | Tiger | | Bird | | Head | | Monarch | |
|---|---|---|---|---|---|---|---|---|---|---|---|---|
| (Backbone) | PSNR | SSIM | PSNR | SSIM | PSNR | SSIM | PSNR | SSIM | PSNR | SSIM | PSNR | SSIM |
| DnCNN | 27.43 | 0.88 | 26.24 | 0.89 | 23.75 | **0.84** | 30.67 | 0.93 | 29.87 | **0.87** | 28.26 | 0.93 |
| FFDNet | 19.81 | 0.49 | 25.03 | 0.84 | 19.62 | 0.62 | 16.72 | 0.55 | 26.50 | 0.72 | 28.09 | 0.92 |
| UNet | 27.06 | 0.88 | 26.17 | **0.90** | 23.72 | **0.84** | 31.06 | **0.94** | 29.90 | **0.87** | 28.49 | **0.94** |
| ★ | **27.56** | **0.89** | **26.30** | **0.90** | **23.76** | **0.84** | **31.07** | **0.94** | **29.92** | **0.87** | **28.55** | **0.94** |

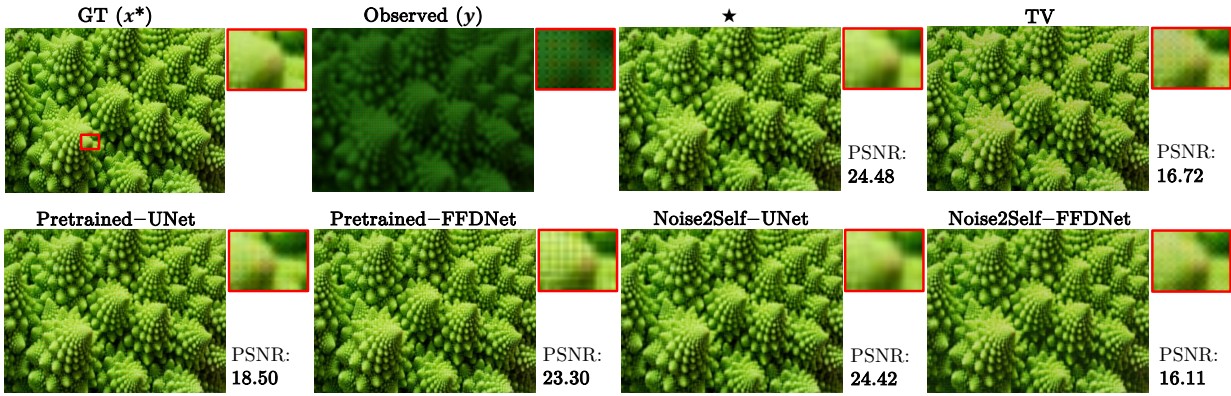

Figure 5: The comparative visualisation of joint deconvolution and demosaicing tasks on the "Fractal" example. The comparison is made across different denoising priors in Play-and-Play framworks, including Single-Shot deep denoising priors (our proposed prior–★, Noise2Self-UNET, and Noise2Self-FFDNet), pre-trained deep denoising priors (Pretrained-UNet and Pretrained-FFDNet), and classical denoising priors (TV).

↻ **Image Deconvolution.** This is a computational technique aimed at reversing the effects of blur on photographs. Mathematically, the observed image, $y$, is the result of convoluing the true image, $x$, with a forward measurement operator: $A(x) = k \otimes x$. The goal of deconvolution is to estimate the original image $x$ by deconvoluting the observed image $y$ with the gaussian kernel $k$. Here, the kernel size was set as 15 with standard deviation 5 of Gaussian distribution.

The performance comparison of various Single-Shot deep denoiser algorithms on a deconvolution task, shown in Table 2, evaluated using both PSNR and SSIM metrics as well. The comparison includes our method alongside established techniques like DnCNN, FFDNet, and UNet across other six image categories. Our approach consistently achieves competitive PSNR scores, surpassing others in the 'Turtle' and 'Monarch' categories, and still shows parity improvements in SSIM values. This indicates not only enhanced accuracy in image reconstruction but also improved perceptual quality. These results underscore our method's effectiveness in noise reduction and sharpness, demonstrating its potential for practical deconvolution applications.

In Figure 4, we illustrate a qualitative comparison of our deconvolution algorithm on a 'Giraffe' image against four leading Single-Shot deep denoising priors. The visual fidelity of our method is apparent, particularly in its capacity to reconstruct intricate details, such as the giraffe's fur that reflected in the zoomed views. This attention to detail extends to the preservation of edge sharpness and the subtle gradations, which contribute to a more natural and cohesive image composition. Contrastingly, other methods exhibit varying degrees of blurring and artifact introduction. This qualitative advancement underscores our algorithm's potential to set a remarkable benchmark for image deconvolution.

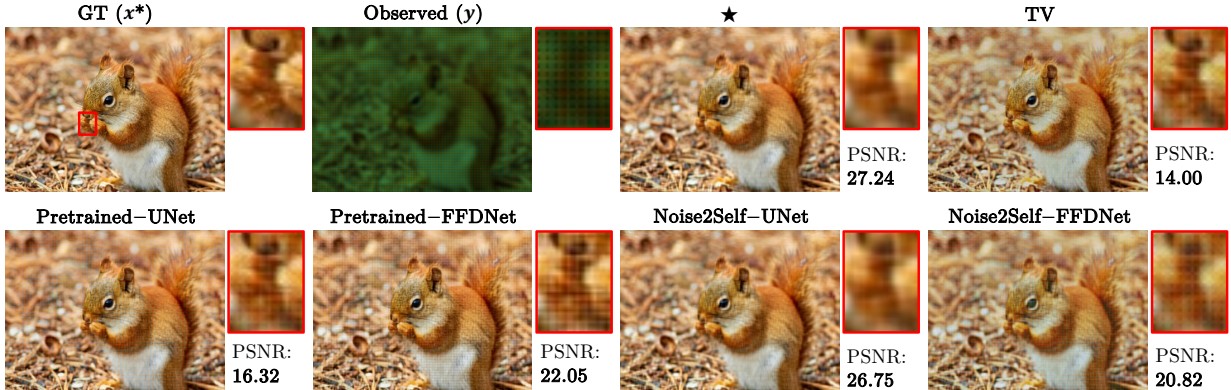

Figure 6: The visualisation comparison of joint deconvolution and demosaicing task on "Squirrel" example between Single-Shot deep denoising priors (our proposed prior–★, Noise2Self-UNET and Noise2Self-FFDNet), pre-trained deep denoising priors (Pretrained-UNet and Pretrained-FFDNet), and classical denoising prior (TV) for denoising in Plug-and-Play algorithm.

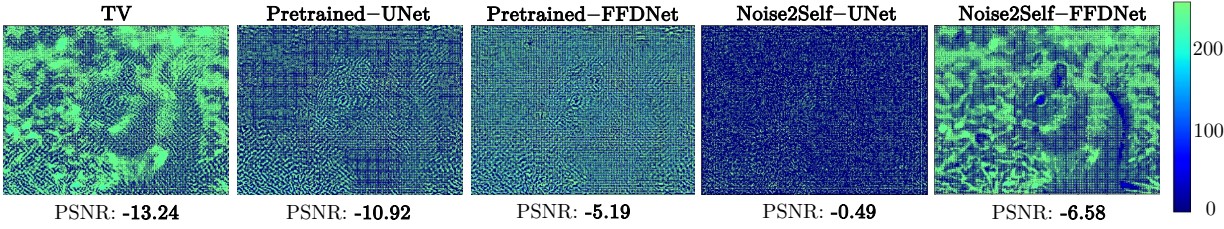

Figure 7: The difference of the error maps between the proposed Single-Shot deep denoising prior –★ and the other compared methods in Figure 6 on the joint deconvolution and demosaicing task on "Squirrel" example.

↻ **Joint Demosaicing and Deconvolution.** Demosaicing is an algorithmic process that reconstructs a full-colour image from the incomplete colour samples output by an image sensor overlaid with a colour filter array (CFA).

The joint demosaicing and deconvolutional process can be represented as: $A(x) = k \otimes (M \odot x)$, where $x$ is the full-colour image, $M$ is the colour filter array (CFA) , $\odot$ denotes element-wise multiplication. The setting of $k$ was same with kernel in deconvolution task.

Based on performances from previous tasks, we note that FFDNet and UNet produce good results. In the more challenging tasks as it is joint demosaicing and deconvolution, these methods, along with classical Total Variation (TV) (Jordan, 1881), are further compared with the data-driven pre-trained denoising priors, and Single-Shot denoising priors training under Noise2Self strategy against our proposed method.

Table 3 illustrates that our method outperforms all methods across all categories, achieving the highest PSNR and SSIM scores in most cases. Notably, in the 'Wolf' and 'Monarch' categories, our method significantly leads, reflecting a substantial improvement in both reconstruction accuracy and image quality, as indicated by the PSNR and SSIM metrics respectively. This demonstrates the effectiveness of our approach in handling complex image restoration tasks.

In Figure 5 and 6, our method outperforms other established algorithms in joint deconvolution and demosaicing, particularly in mitigating chromatic aberrations such as red grids or spots. These artifacts, which degrade image quality, are significantly reduced in our approach, leading to a visually coherent result that aligns more closely with the ground truth. Competing methods, including pre-trained networks and classical denoising techniques, often introduce or inadequately suppress such distortions, resulting in inferior colour and contrasts outcomes. Our method advances the visual quality of image restoration, effectively preserving the natural colour and detail fidelity.

Table 3: The performance (PSNR(dB), SSIM) comparison on joint deconvolution and demosaicing of Single-Shot deep denoising priors (★, Noise2Self-UNET, and Noise2Self-FFDNet), pre-trained deep denoising priors following (Lai et al., 2023) (Pretrained-UNet and Pretrained-FFDNet), and classical denoising priors (TV) in Plug-and-Play framework. ★ denotes our proposed denoising prior in the SS-PnP strategy.

|  |  | Wolf | | Beach | | Mushroom | | Bird | | Head | | Monarch | |
|---|---|---|---|---|---|---|---|---|---|---|---|---|---|
|  |  | PSNR | SSIM | PSNR | SSIM | PSNR | SSIM | PSNR | SSIM | PSNR | SSIM | PSNR | SSIM |
| Classic | TV | 17.27 | 0.45 | 16.69 | 0.48 | 17.83 | 0.53 | 18.72 | 0.63 | 17.53 | 0.58 | 16.35 | 0.41 |
| Pretrain | UNet | 17.54 | 0.36 | 18.02 | 0.48 | 18.09 | 0.49 | 19.54 | 0.65 | 19.12 | 0.57 | 17.10 | 0.41 |
|  | FFDNet | 25.66 | 0.81 | 24.06 | 0.75 | 25.02 | 0.82 | 24.30 | 0.82 | 24.65 | 0.77 | 21.82 | 0.63 |
| SS-PnP | UNet | 30.54 | 0.90 | 25.52 | 0.77 | 26.86 | 0.87 | 27.75 | 0.87 | 27.96 | 0.81 | 26.31 | **0.89** |
|  | FFDNet | 18.45 | 0.28 | 16.17 | 0.25 | 18.78 | 0.34 | 18.15 | 0.45 | 18.68 | 0.29 | 15.39 | 0.26 |
|  | ★ | **30.90** | **0.91** | **27.37** | **0.85** | **27.00** | **0.88** | **28.20** | **0.89** | **28.02** | **0.82** | **26.48** | **0.89** |

We also present a visualisation comparing the error maps of our proposed prior with other denoising priors within the Plug-and-Play framework in Figure 7. While there are some noticeable visual differences, the distribution of errors across the image suggests that many discrepancies are not easily detectable by the human eye. However, our method demonstrates a significant numerical improvement over the alternatives, despite these subtle visual distinctions.

### 4.3 Ablation study

We provide a further empirical study for comparing our proposed implicit neural network (INR) with the existing classical INR, SIREN (Strümpler et al., 2022).

We follow the same experimental settings as described in 4.1, and measure the performance based on Peak Signal-to-Noise Ratio (PSNR). In Table 4, our proposed INR outperforms SIREN in all the 4 tasks: deconvolution, super resolution 2× and 4×, and joint deconvolution and demosaicing. This performance is significant in the super resolution with an average improvement of 0.8dB. The spatial compactness brought by our INR guarantees the representativeness of different levels of feature that pushes the performance of super resolution tasks.

We also performed a comparison with other classical implicit neural representation (INR) methods, such as WIRE (Saragadam et al., 2023). However, WIRE showed inconsistent performance and lacked stability across the various tasks. In the Bird example, our method demonstrated superior PSNR results compared

Table 4: The performance (PSNR(dB)) comparison of the implicit neural priors on deconvolution (Deconv), super resolution (SR) with 2× and 4× upscalings, and joint deconvolution and demosaicing (Joint) tasks. The ★ denotes our proposed implicit neural prior, which we compared with the SOTA implicit neural representation, SIREN.

| Example | INR Prior | **Deconv** | **SR (2×)** | **SR (4×)** | **Joint** |
|---|---|---|---|---|---|
| **Wolf** | SIREN | 33.05 | 26.08 | 22.26 | 30.83 |
|  | ★ | **33.07** | **27.21** | **23.30** | **30.90** |
| **Bird** | SIREN | 30.71 | 25.72 | 21.83 | 28.13 |
|  | ★ | **31.07** | **26.44** | **22.06** | **28.20** |
| **Head** | SIREN | 29.82 | 23.69 | 19.57 | 27.79 |
|  | ★ | **29.92** | **24.24** | **20.61** | **28.02** |

Table 5: The performance (PSNR(dB)) comparison of BM3D (Dabov et al., 2007), DIP (Sun et al., 2021b) and our proposed ★ priors on super-resolution (SR) task with 2× and 4× upscalings in the Plug-and-Play framework.

| PnP | Fractal | | Tiger | | Bird | |
|---|---|---|---|---|---|---|
| Framework | 2× | 4× | 2× | 4× | 2× | 4× |
| BM3D | 12.62 | 12.35 | 15.59 | 14.66 | 11.89 | 11.46 |
| DIP | 22.92 | 18.51 | 22.14 | 18.34 | 23.79 | 20.49 |
| ★ | **25.43** | **21.26** | **23.12** | **19.45** | **26.44** | **22.06** |

to WIRE, with 26.44dB compared to 24.88dB for 2× super-resolution, 22.06dB compared to 20.59dB for 4× super-resolution, and 28.20dB compared to 27.83dB for the joint deconvolution and demosaicing task.

We can observe from Figure 8 that our INR excels in reconstructing the image with smoother edge features and enhanced colour representation. The difference is more clearly reflected in the error maps, with a noticeable variation around the edge features across all four tasks. The performance over iteration curves demonstrate our leading advances over the optimisation iterations, meanwhile show the early iteration steps is effective for achieving the best performance. This is commonly observed in iterative optimisation techniques (Wei et al., 2020; 2022), where initial iterations tend to improve the quality of the reconstruction, while beyond a certain point, the model begin to introduce artifacts or oversmooth the image, which leads to a degradation in performance and consequently a drop in PSNR.

We also conducted a comparative analysis of our proposed Single-Shot prior against other state-of-the-art priors within Plug-and-Play frameworks, as shown in Figure 5. The BM3D (Dabov et al., 2007) and DIP (Sun et al., 2021b) priors were each run for 240 iterations—ten times the iterations used for our Single-Shot approach. Despite this, the classic BM3D method's performance was markedly lower compared to both DIP and our method. While the DIP prior within the Plug-and-Play framework demonstrated results closer to ours, a significant performance gap remains, underscoring the superior efficacy of our approach in both 2× and 4× super-resolution tasks.

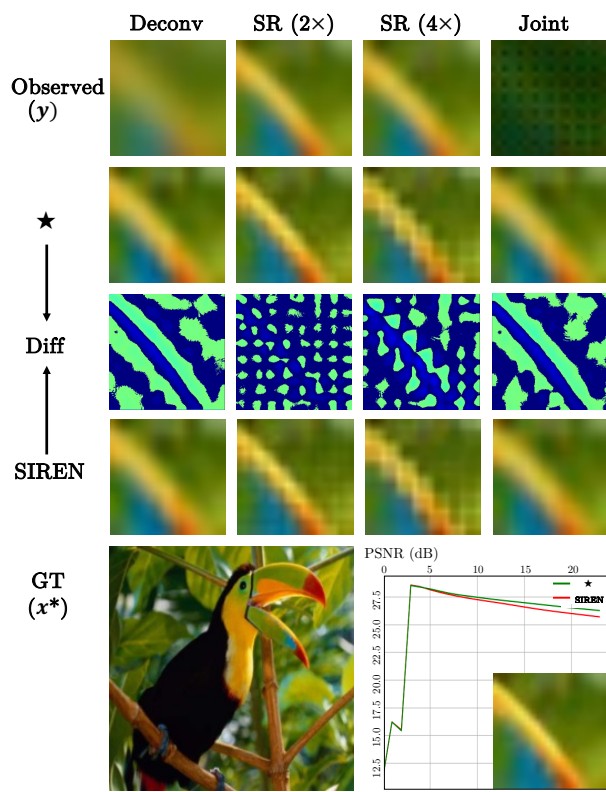

Figure 8: The visualisation comparison of the implicit neural priors (Ours–★ and SIREN) on deconvolution (Deconv), super resolution(SR) with 2× and 4× upscalings, and joint deconvolution and demosaicing (Joint) tasks on "Bird" example. The difference of the error maps of ★ and SIREN are provided in the third row. The plot indicates the performance (PSNR) with both implicit neural priors changing over the ADMM iteration on the joint deconvolution and demosaicing task.

## 5   Conclusion

Our work pioneers the Single-Shot Plug-and-Play (SS-PnP) method, transforming the use of PnP priors in inverse problem-solving by reducing reliance on large-scale pre-training of denoisers, and using a single instance. We also propose Single-Shot proximal denoisers via implicit neural priors allowing for superior approximation quality for solving inverse problems using only a single instance. We propose a novel function for implicit neural priors that has desirable theoretical guarantees. We also showed that our work generalises well across different tasks, showing empirical stability for single and multiple operators. We then open the door to a new research line for Single-Shot PnP. Whilst this work uses mainly PnP-ADMM due to the well-know properties, and performance in comparison with other algorithms. Future work will include to evaluate our new research line on Single-Shot PnP on different algorithms including but not limited to FISTA (Beck & Teboulle, 2009a), HQS (Geman & Yang, 1995), and Primal-dual (Dantzig et al., 1956) algorithms. Another additional side insight could be to explore the result over randomising $z^0$ in step 2 of Algorithm 1. Though the primary focus of this work is not the convergence of the proposed implicit neural prior, the convergence of the broader PnP framework is a significant and intricate topic that is valuable to analysis in the future research work.

## Acknowledgements

This project was supported with funding from the Cambridge Centre for Data-Driven Discovery and Accelerate Programme for Scientific Discovery, made possible by a donation from Schmidt Futures. YC is funded by an AstraZeneca studentship and a Google studentship. The work of RHC was partially supported by HKRGC GRF grants CityU11309922, CRF grant C1013-21GF and HKITF MHKJFS Grant MHP/054/22. CBS acknowledges support from the Philip Leverhulme Prize, the Royal Society Wolfson Fellowship, the EPSRC advanced career fellowship EP/V029428/1, EPSRC grants EP/S026045/1 and EP/T003553/1, EP/N014588/1, EP/T017961/1, the Wellcome Innovator Awards 215733/Z/19/Z and 221633/Z/20/Z, the European Union Horizon 2020 research and innovation programme under the Marie Skodowska-Curie grant agreement No. 777826 NoMADS, the Cantab Capital Institute for the Mathematics of Information and the Alan Turing Institute. AIAR acknowledges support from CMIH (EP/T017961/1) and CCIMI, University of Cambridge. This work was supported in part by Oracle Cloud credits and related resources provided by Oracle for Research. Also, EPSRC Digital Core Capability.

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
