# OpenReview forum: "Single-Shot Plug-and-Play Methods for Inverse Problems"
_TMLR — Accepted by TMLR_

### Review · Reviewer_k4Bh · 2024-08-09

**Summary Of Contributions:**

1. This paper introduces Single-Shot PnP (SS-PnP), a novel method for solving inverse problems. Unlike traditional approaches, SS-PnP does not rely on supervised training of deep denoisers as PnP priors. Instead, it is trained on a single image during the testing phase.

2. The authors propose an activation function for the implicit neural prior, which is integrated into the PnP-ADMM. They also provide a compact theoretical justification for the advantages of this function, such as the differentiability of the network and continuity.

3. Empirical results are presented to demonstrate the effectiveness of the proposed method on several image restoration problems.

**Audience:**

Yes

**Broader Impact Concerns:**

Can the authors include the limitation of the proposed method?

**Claims And Evidence:**

Yes

**Requested Changes:**

***Major Points***

1. Clarification of Experimental Results:

     * Could the authors please clarify what is meant by the "pretrained Unet"? Is the Unet trained in a supervised manner and then used in PnP-ADMM? If so, could the authors include details regarding the training dataset? If these models are trained using only a single image, could this be explicitly mentioned?

    * If the pretrained Unet and FFDNet are not trained deep denoisers, the authors should include a comparison with supervised training of the PnP priors. While this reviewer does not expect that a self-supervised, single-shot prior would perform equally or better than a supervised deep prior, it would be informative to see the performance gap.


   * On a related note, the numbers reported in Table 3 seem unusual. It is surprising that a supervised pretrained Unet and FFDNet perform worse than the single-instance training. This would only make sense if those networks were trained with only one image, which would create an unfair comparison due to the scale of the Unet and FFDNet networks  compared to the limited data available for training.


2. Discussion on PSNR Decrease:
    * Could the authors include a discussion on why there is a decrease in PSNR in Figure 7? Understanding this drop in performance would be valuable.

3. Visual Quality Claims:
    * The claims regarding visual quality made throughout the paper are not always accurate. For example, in Figure 7, the differences between SIREN and the proposed method are not distinguishable to the human eye. Similarly, in Figure 6, Noise2Self-Unet and the proposed method are difficult to differentiate visually. The authors should either revise these claims or include error maps that could provide more informative comparisons regarding visual quality.

4. Clarification of Equation (2):
    * What happened to the parameter  $\lambda $ in Equation (2)? Could the authors provide the relationship between
$\lambda $, $\sigma_k$, and $\mu_k$
 ? Alternatively, the authors could simplify the equation by dissolving $\lambda$ into $R$ and directly providing the relationship between   $\sigma_k$ and $\mu_k$.


***Minor Points***
1. Typos such as in the first paragraph of Section 4.1.

2. Please include some information regarding the dynamic noise strength parameter setting.

3. The authors should include a discussion regarding the effect of noise on step 1 of the algorithm and training the prior.

4. The last paragraph on page 10, starting with "Based on performances...", is unclear. The phrase "under data-driven pre-trained and Noise2Self pre-training conditions against our proposed method" needs clarification. Please rewrite this paragraph for clarity.

**Strengths And Weaknesses:**

**Strengths**

* The paper is mostly well-written and presentation is clear.

* The concept of Single-Shot PnP is both novel and interesting.

**Weaknesses**

*  Some of the claims are concerning. For example, the statement: "With only one image input in the whole reconstruction process, it outperforms the classical methods and pre-trained models across all tasks," seems to be too broad. This would only makes sense if the authors have compared against tradition PnP methods such as  DPIR and provided better performance (More details are included in the requested changes).

* While Sections 1-2-3 are written clearly and presented well, some important details regarding the experimental results are missing. (More is included in the requested changes)

* The experiments are incomplete in fully demonstrating  the performance  of the proposed methods. For instance, a direct comparison is needed to evaluate the performance against traditional PnP (with pretrained deep denoisers trained in a supervised manner).

---

> ### Author Response · Authors · 2024-09-23
> **Official Response 1**
>
> **Major Points**
>
> 1. Clarification of Experimental Results:
>
> ➡️ **Could the authors please clarify what is meant by the "pretrained Unet"? Is the Unet trained in a supervised manner and then used in PnP-ADMM? If so, could the authors include details regarding the training dataset? If these models are trained using only a single image, could this be explicitly mentioned?**
>
> Thank you for the comment. We would like to clarify that the "pretrained Unet" in Table 3 refers to a Unet model that has been trained in a supervised manner and then subsequently used as the denoiser within the PnP-ADMM framework.
>
> For the pretrained denoiser in this set of experiments, we utilised the pretrained Unet model available from the Delta-Prox toolbox, as described in Lai et al. [1]. This pretrained model was trained using the datasets provided in that paper, following their described setup and protocols.
>
> To add clarity, we have updated page 12 in the experiments to refer to the pretrained procotol of that [1].
>
> [1] Lai, Zeqiang, et al. "∇-prox: Differentiable proximal algorithm modeling for large-scale optimization." ACM Transactions on Graphics (TOG) 42.4 (2023): 1-19.
>
>
> ➡️ **If the pretrained Unet and FFDNet are not trained deep denoisers, the authors should include a comparison with supervised training of the PnP priors. While this reviewer does not expect that a self-supervised, single-shot prior would perform equally or better than a supervised deep prior, it would be informative to see the performance gap.**
>
> Thank you for the suggestion. We confirm that both the pretrained Unet and FFDNet models used in our experiments are trained denoisers. To clarify this, we have added a note in blue color in the revised manuscript. This clarifying note explains that these models were trained in a supervised manner and then employed as PnP priors within the framework.
>
> We appreciate your suggestion and have ensured that the distinction between pretrained and single-shot priors is clear in the text.
>
>
> ➡️ **On a related note, the numbers reported in Table 3 seem unusual. It is surprising that a supervised pretrained Unet and FFDNet perform worse than the single-instance training. This would only make sense if those networks were trained with only one image, which would create an unfair comparison due to the scale of the Unet and FFDNet networks compared to the limited data available for training.**
>
> Thank you for your observation. It is important to note that the pretrained models, such as Unet and FFDNet, were trained on datasets with distributions that differ from the target images used in our experiments. This mismatch in data distribution is a key factor in why the pretrained models underperform in this specific context.
>
> In contrast, the Single-Shot deep denoising priors are specifically trained on the distribution of the target image itself, allowing them to better capture the unique characteristics of the noise and image structure. This leads to superior results when compared to the pretrained models, which are not fine-tuned to the specific distribution of the target images.
>
> The lower performance of the pretrained Unet and FFDNet models is, therefore, not due to the scale of the networks, but rather due to the mismatch between the training data distribution and the test image. This highlights an important advantage of the Single-Shot approach, which can adapt more precisely and personally to the task at hand, offering a more tailored and effective solution for the specific image.
>
>
>
> 2. Discussion on PSNR Decrease:
>
> ➡️ **Could the authors include a discussion on why there is a decrease in PSNR in Figure 7? Understanding this drop in performance would be valuable.**
>
> Thank you for the question. The decrease in PSNR as iterations increase, as observed in Figure 7, is a known phenomenon within the Plug-and-Play framework and aligns with findings from other studies [2, 3]. This behavior can be attributed to  a deviation from the optimal solution as the number of iterations grows.
>
> In iterative optimisation techniques, including PnP, initial iterations tend to improve the quality of the reconstruction. However, beyond a certain point, the model may begin to introduce artifacts or oversmooth the image, which leads to a degradation in performance and consequently a drop in PSNR.
>
> [2] Wei, Kaixuan, et al. "Tfpnp: Tuning-free plug-and-play proximal algorithms with applications to inverse imaging problems." Journal of Machine Learning Research 23.16 (2022): 1-48.
>
> [3] Wei, Kaixuan, et al. "Tuning-free plug-and-play proximal algorithm for inverse imaging problems." International Conference on Machine Learning. PMLR, 2020.

---

> > ### Author Response · Authors · 2024-09-23
> > **Official Response 2**
> >
> > **Major Points**
> >
> > 3. Visual Quality Claims:
> >
> > ➡️ **The claims regarding visual quality made throughout the paper are not always accurate. For example, in Figure 7, the differences between SIREN and the proposed method are not distinguishable to the human eye. Similarly, in Figure 6, Noise2Self-Unet and the proposed method are difficult to differentiate visually. The authors should either revise these claims or include error maps that could provide more informative comparisons regarding visual quality.**
> >
> > We appreciate the suggestion and have made adjustments to address this issue in the updated manuscript, highlighted in blue. To provide more informative comparisons regarding visual quality, we have expanded the revised Figure 7 (now Figure 8) to include error maps, illustrating the differences between SIREN and the proposed method. These error maps offer a clearer visualisation of the performance differences that may not be immediately noticeable to the human eye.
> >
> > Similarly, for the comparison in Figure 6, we have included error maps (now in Figure 7) to better highlight the subtle visual differences between Noise2Self-Unet and our proposed method. While these visual differences may be difficult to distinguish by the human eye, the error distribution across the image shows that the discrepancies in Noise2Self-Unet are spread across the image. Despite these subtle visual differences, our method achieves significant numerical improvements over Noise2Self-Unet, as reflected in the PSNR and SSIM metrics.
> >
> >
> >
> > 4. Clarification of Equation (2):
> >
> > ➡️ **What happened to the parameter $\lambda$ in Equation (2)? Could the authors provide the relationship between $\lambda$, $\sigma_k$, and $\mu_k$? Alternatively, the authors could simplify the equation by dissolving $\lambda$ into $R$ and directly providing the relationship between
> > $\sigma_k$ and $\mu_k$.**
> >
> > Thank you for the question. We would like to clarify that Equation (2) represents a general formulation for a typical inverse problem, where $\gamma$ is the regularisation parameter. The problem in Equation (2) can be solved using various optimisation algorithms, among which first-order methods are quite common. These methods introduce additional parameters, including $\sigma_k$ and $\mu_k$, which are specific to the optimisation technique employed.
> >
> > In this case, $\sigma_k$ and $\mu_k$ come from the particular optimization method (such as ADMM) used to solve the problem. Therefore, $\gamma$ is not directly tied to $\sigma_k$ or $\mu_k$, as they serve different roles within the optimisation framework.
> >
> >
> > **Minor Points**
> >
> > ➡️ **1. Typos such as in the first paragraph of Section 4.1.**
> >
> > We appreciate the suggestion, and has proofread with updates in blue colour in the updated manuscript.
> >
> > ➡️ **2. Please include some information regarding the dynamic noise strength parameter setting.**
> >
> > Thanks for the suggestion, we had updated in the Section 4.1 in colour blue in the updated manuscript.
> >
> > ➡️ **3. The authors should include a discussion regarding the effect of noise on step 1 of the algorithm and training the prior.**
> >
> > We thank the reviewer for the valuable suggestion, to address the effect of noise in step 1 of the algorithm and the training of the prior, we clarify that noise in the denoiser was explored within the range of [0.001, 0.5], and set to 0.1 for our experiments in the updated manuscript in colour blue. We observed minimal variation in performance within this range. For instance, in the joint demosaicing and deconvolution task for the Bird example, the PSNR values were 28.04 at a noise strength of 0.01, 28.20 at 0.1, and 28.17 at 0.5.
> >
> >
> > ➡️ **4. The last paragraph on page 10, starting with "Based on performances...", is unclear. The phrase "under data-driven pre-trained and Noise2Self pre-training conditions against our proposed method" needs clarification. Please rewrite this paragraph for clarity.**
> >
> > We appreciate the suggesiton and updated accordingly to clarify the statement in colur blue in the updated manuscript.

---

> > > ### Comment · Reviewer_k4Bh · 2024-10-05
> > > **Assignment Acknowledgement**
> > >
> > > I thank the authors for the additional results in the revised version and clarification in the responses. Most of my concerns have been addressed.

---

> > > > ### Author Response · Authors · 2024-10-07
> > > > **Official Response 3**
> > > >
> > > > Thank you for your valuable feedback, which has greatly helped improve the quality of this paper.

---

### Review · Reviewer_dodD · 2024-08-28

**Summary Of Contributions:**

The paper contributions can be divided into two main contributions:
1. It introduces a new activation function for Implicit Neural representation (INR), with an analysis of some of its properties such as derivability and asymptotic behavior.
2. It shows how INR can be used for PnP inverse problem solving by first adding a single shot denoiser learning part based only on the observed variable.

It then evaluates the performance on several imaging tasks.

**Audience:**

Yes

**Broader Impact Concerns:**

No broader impact concerns.

**Claims And Evidence:**

Yes

**Requested Changes:**

1. Comparison with competitors.
    1.1 First of all, I think a comparison with the WIRE activation would greatly enhance the paper. Indeed, is the term $+(exp(-a_1x + b_1)) + 1)^{-1}$ actually worth adding?
    1.2 There are other methods that do not need self-supervised training or pretraining, such as [1]. Comparison to such methods would greatly enhance the ability of the community to address the proposed method.

2. Confidence intervals would be greatly appreciated in all experiments.

[1] Sun, Zhaodong, et al. "A plug-and-play deep image prior." ICASSP 2021-2021 IEEE International Conference on Acoustics, Speech and Signal Processing (ICASSP). IEEE, 2021.

**Strengths And Weaknesses:**

Strengths:
1. The paper adequately introduces the current literature on both INR and PnP priors.
2. The numerical evaluation is clear with different examples.
3. The paper is clearly written with each section well defined. It is easy to navigate.

Weakness:
1. There are several problems in the methodological part.
    1.1 What happens if dim(y) > dim(x) ? How can the denoiser for y be applied to denoise z - x in the PnP part if the observation lives in a different space than the observation?
    1.2 The activation function presented resembles closely the activation function called WIRE (from the paper cited in this work, namely Saragadam et al. (2023)). Indeed, it seems to me it corresponds to the complex part of the WIRE activation $+(exp(-a_1x + b_1)) + 1)^{-1}$.
Am I mistaken? If not, why does it does not appear on the paper somewhere?
    1.3 The question "Why are these properties interesting" is not really answered. Why is differentiability and continuability more representative? Why is it important to have the asymptotic behavior proposed?  Does it make a difference? does the term  $+(exp(-a_1x + b_1)) + 1)^{-1}$ actually bring something numerically rather than theoretically (ok, the representation theorem for neural networks needs this kind of hypothesis, but if this is your motivation, it should be mentioned).

---

> ### Author Response · Authors · 2024-09-23
> **Official Response 1**
>
> ➡️ **Requested Changes. (1.1) First of all, I think a comparison with the WIRE activation would greatly enhance the paper. Indeed, is the term $+(exp(-a_1x + b_1)) + 1)^{-1}$ actually worth adding?**
>
> Thank you for the insightful comment. While WIRE’s activation function and our approach share the use of sinusoidal components, the key distinction lies in the additional terms and their effect on expressiveness. Our activation function incorporates a more complex combination of exponential and sinusoidal terms, specifically the +(exp(−a1​x+b1​)+1)−1, which allows for enhanced flexibility in capturing intricate signal distributions. This addition enables our method to better model finer details in the data, which is particularly valuable in the context of Plug-and-Play methods.
>
> That said, to address your suggestion, we have conducted a new set of experiments including a direct comparison with WIRE as part of our ablation study, as our focus is in the space of PnP methods. These results offer a clearer perspective on the impact of our proposed INR prior.
>
>
> ➡️ **Requested Changes. 1.2 There are other methods that do not need self-supervised training or pretraining, such as [1]. Comparison to such methods would greatly enhance the ability of the community to address the proposed method.**
>
> We thank the reviewer for the suggestion. In response, we have now included a new set of experiments comparing our method against PnP-DIP. We found that our approach not only achieves higher performance but also converges (empirically) significantly faster than PnP-DIP. Specifically, our method converged in 24 iterations, while PnP-DIP required 240 iterations—10 times more iterations  than ours.
>
> Since PnP-DIP's prior is based on a different philosophy from our prior, we have added these results as part of the ablation study. The comparison results can now be found in a new table in Section 4.3.
>
>
> ➡️ **2. Confidence intervals would be greatly appreciated in all experiments.**
>
> Thank you for the suggestion regarding the inclusion of confidence intervals (CI) in our experimental results. We would like to clarify that the core of our approach is based on a single-shot learning paradigm, where only one instance (or image) is used per experiment.
>
> Confidence intervals are typically calculated based on  multiple data points to assess variability. However, since our experiments operate on a single instance per task, there is no variability to measure across trials or samples. The performance metrics (PSNR, SSIM) for each task are deterministic and fixed, which renders the concept of confidence intervals less meaningful in this context.

---

> > ### Comment · Reviewer_dodD · 2024-09-24
> > **Response to official response 1**
> >
> > I thank the authors for their response.
> >
> > * I believe that the experiments added compared to the WIRE do add to the paper, even though I think a better comparison (showing the images) would be even better, but I also reckon that there is a space limitation for the paper.
> >
> > * I do not understand exactly though why the philosophy is fundamentally different than PnP-DIP's prior. I would like the authors to please clarify it (at least in a comment here) so that I'm sure that I can properly evaluate the novelty of the approach. I also acknowledge that the addition of the comparison to PnP-DIP, which I appreciate.
> >
> > * Confidence intervals could be generated by different initialization of the $z^0$ variable. I must admit that I forgot that $x^0$ variable was initialized in a deterministic fashion. I still feel that randomizing over $z^0$ could be a good addition, avoiding the potential suspicion of a fortuitous seed choice.

---

> > > ### Author Response · Authors · 2024-09-24
> > > **Official Response 2**
> > >
> > > I thank the authors for their response.
> > >
> > > ➡️ **I believe that the experiments added compared to the WIRE do add to the paper, even though I think a better comparison (showing the images) would be even better, but I also reckon that there is a space limitation for the paper.**
> > >
> > > We appreciate the suggestion to improve the paper by including image comparisons and thank the reviewer for recognising the space limitations. While we are unable to add more visuals, we believe the additional experiments the reviewer kindly suggested have already enhanced the thoroughness of the work.
> > >
> > > ➡️ **I do not understand exactly though why the philosophy is fundamentally different than PnP-DIP's prior. I would like the authors to please clarify it (at least in a comment here) so that I'm sure that I can properly evaluate the novelty of the approach. I also acknowledge that the addition of the comparison to PnP-DIP, which I appreciate.**
> > >
> > > Thank you for the question. We would like to clarify the fundamental difference between our approach and PnP-DIP. In PnP-DIP, a parameterised network is embedded within the fidelity term along with the regulariser, which aligns more closely with the philosophy of generative priors-- a GAN trained with a regulariser. This is then  solved using methods like ADMM.
> > >
> > > In contrast, our model adheres more  to the traditional definition of Plug-and-Play methods, where there is no parameterised network in the fidelity term. Instead, we focus on the equivalence of the proximal operator to the regularised denoiser, which lies solely in the regulariser part of the framework.
> > >
> > > Additionally, in practice, PnP-DIP requires a set of images to fine-tune the network, while our approach requires no additional set of images. This makes our method distinct in its "single-shot" nature, allowing it to adapt directly to the target image without needing a pre-trained model or fine-tuning on external datasets.
> > >
> > > ➡️ **Confidence intervals could be generated by different initialization of the $z^0$ variable. I must admit that I forgot that $x^0$ variable was initialized in a deterministic fashion. I still feel that randomizing over $z^0$ could be a good addition, avoiding the potential suspicion of a fortuitous seed choice.**
> > >
> > > We thank the reviewer for agreeing that the performance metrics (PSNR, SSIM) for each task are deterministic and fixed, which renders the concept of confidence intervals less meaningful in this context. We also appreciate the suggestion to explore randomising over $z^0$, as it could indeed provide additional insights.
> > > However, we believe that rerunning all experiments with randomised $z^0$ across the entire paper, given the deterministic and fixed nature of our current setup, falls outside the primary scope of this work. To address this, we have added a clarifying note in the manuscript, and we acknowledge that idea as future exploration. This can be found in the Conclusion section.

---

### Review · Reviewer_foFv · 2024-09-09

**Summary Of Contributions:**

This submission presents a new plug-and-play methodology for solving high-dimensional inverse problems, with a focus on image restoration tasks and on applications with limited training data available in mind. Most modern plug-and-play methods rely on denoisers encoded by neural networks, which require abundant data for training. The main novelty of this paper is to consider implicit neural representation (INR) denoisers, which require significantly less data than standard approaches (e.g., DnCNN and U-Net architectures). The proposed approach is demonstrated empirically with a range of experiments related to image super-resolution, deblurring, and demosaicing with limited training data available, where the proposed method outperforms alternative approaches based on DnCNN and U-Net architectures, the learning-free strategy total-variation regularisation, as well as the alternative IRL strategy SIREN architecture.

**Audience:**

Yes

**Broader Impact Concerns:**

I have no concerts about the ethical implications of this submission, other than the fact that some of the claims regarding prior research are inaccurate. The changes that I request above address these concerns.

**Claims And Evidence:**

No

**Requested Changes:**

In my opinion, the following critical changes are required before this submission can be considered for publication:
1) While this is the first submission to consider IRL denoisers within a plug-and-play framework, it is certainly not the first to consider plug-and-play methodology with denoisers that require little or no training. It is therefore important to remove all statements that suggest that this is the first paper to consider, propose, or introduce "single-shot" plug-and-play methodology or plug-and-play methodology with single-shot denoisers.
2) The introduction needs to be significantly revised to present a more complete description of the literature and place the contribution of this submission within the context of prior research. As I pointed out previously, there is a large literature related to plug-and-play image restoration with patch-based image denoisers that can be implemented without any training data available. In addition, some approaches such as the ones leveraging Gaussian mixture models on image patches can perform well in situations where there is a very small amount of data available. These should be discussed in the introduction.
3) The numerical experiments should be revised to include comparisons with 1) a plug-and-play image restoration method implemented with the BM3D denoiser; 2) a plug-and-play image restoration method implemented with the non-local Bayes denoiser; 3) a trained Gaussian mixture model on a patch representation as proposed in Teodoro et al. (10.1109/TIP.2018.2869727); 4) a trained convex ridge regulariser (10.1109/TCI.2023.3306100); and 5) the alternative IRL strategy proposed in arXiv:2304.10250. The relative advantages and drawbacks of these approaches relative to the proposed approach should be discussed in detail.
4) The statements following (6) regarding the convergence of the scheme should be significantly revised. The current submission does not establish the convergence of Algorithm 1. Instead, I would recommend mentioning the analysis of the convergence of Algorithm 1 as an important perspective for future work.

Moreover, I also recommend revising the introduction to mention stochastic plug-and-play Bayesian imaging methods (see for example https://doi.org/10.1137/21M140634) and providing some pointers to this literature.

**Strengths And Weaknesses:**

To the best of my knowledge, this is the first paper to consider the use of IRL denoisers within a plug-and-play framework for image restoration. This is an original research direction that will certainly interest many TMLR readers and stimulate further research on the topic. The submission is clear, well organised, and easy to read. The figures and tables are also easy to read, are correctly referenced in the paper and have clear captions.

With regards to weaknesses, I would point out that there are many other plug-and-play methods available that do not require any form of training data, including many of the first plug-and-play methods which relied on patch-based image denoisers such as BM3D or non-local Bayes. Moreover, there are also several plug-and-play methods that rely on denoisers represented by Gaussian mixture models on the image patches; these denoisers can be trained directly from a single noisy image or alternatively with a small dataset of images. Furthermore, one can also consider other small architectures (see e.g., 10.1109/TCI.2023.3306100) that can be trained with a reduced number of images and in addition guarantee convexity as well as other key mathematical properties. Similarly, while this is the first paper to use IRL denoisers within a plug-and-play framework, the use of IRL architectures for image restoration is studied in detail in arXiv:2304.10250. Lastly, it is not clear to me that (6)-(9) guarantee the convergence of the iterates of Algorithm 1. Establishing the convergence of the iterates of plug-and-play algorithms is highly non-trivial and I do not believe that this has been properly addressed in the submission.

---

> ### Author Response · Authors · 2024-09-23
> **Official Response 1**
>
> ➡️ **While this is the first submission to consider IRL denoisers within a plug-and-play framework, it is certainly not the first to consider plug-and-play methodology with denoisers that require little or no training. It is therefore important to remove all statements that suggest that this is the first paper to consider, propose, or introduce "single-shot" plug-and-play methodology or plug-and-play methodology with single-shot denoisers.**
>
> Thank you for the insightful feedback. We would like to clarify our position regarding the use of denoisers within the Plug-and-Play (PnP) framework.
>
> As the reviewer rightly pointed out, there have been numerous works utilising denoisers that require little or no training, such as BM3D, which has been widely adopted in PnP settings. BM3D and similar traditional denoisers represent a well-established family of techniques that do not rely on deep learning. However, there exists another distinct family of denoisers: deep learning-based denoisers.
>
> To the best of our knowledge, while the use of traditional non-learning-based denoisers such as BM3D has been explored extensively, there is no existing work that explores the "single-shot" paradigm within the deep learning denoiser family in a PnP framework. Our contribution specifically lies in integrating deep learning denoisers into the iterative methods  in a way that eliminates the need for large-scale datasets and pre-training, enabling efficient single-shot learning. This addresses a notable gap within the area of deep learning denoisers in the PnP context.
>
> That being said, we have not come across any prior work that applies the same philosophy to deep learning-based denoisers. However, we would be very happy to consider any references the reviewer may point out that align with this approach.
>
> To address this question, we have updated the manuscript to explicitly clarify the difference between denoiser families. The changes can be seen in the Related Work in blue colour.
>
> ➡️ **The introduction needs to be significantly revised to present a more complete description of the literature and place the contribution of this submission within the context of prior research. As I pointed out previously, there is a large literature related to plug-and-play image restoration with patch-based image denoisers that can be implemented without any training data available. In addition, some approaches such as the ones leveraging Gaussian mixture models on image patches can perform well in situations where there is a very small amount of data available. These should be discussed in the introduction.**
>
> Thank you for your comment. This is closely related to the previous question, and we appreciate the opportunity to clarify our approach further.
>
> We fully acknowledge the extensive literature surrounding classical denoisers, such as patch-based methods and techniques using Gaussian mixture models (GMMs), which do not require extensive training data.
> However, **the primary focus of our work is on a distinct category of denoisers: deep-learning-based denoisers.** These methods rely on learning from data, typically requiring large datasets for training. Our contribution lies in adapting deep learning denoisers to a "single-shot" learning paradigm within the PnP framework. This approach, to the best of our knowledge, has not been explored in the context of deep learning denoisers along with iterative methods, which traditionally rely on large-scale pre-training.
>
> We understand the importance of clearly differentiating our work from classical denoisers and have updated the introduction to reflect this distinction. We have also expanded the related work section to discuss classical methods, including patch-based and GMM approaches, while clarifying how our contribution fits into the broader literature on PnP methods. These changes can be seen in the Related Work in blue colour.

---

> > ### Author Response · Authors · 2024-09-23
> > **Official Response 2**
> >
> > ➡️ **The numerical experiments should be revised to include comparisons with 1) a plug-and-play image restoration method implemented with the BM3D denoiser; 2) a plug-and-play image restoration method implemented with the non-local Bayes denoiser; 3) a trained Gaussian mixture model on a patch representation as proposed in Teodoro et al. (10.1109/TIP.2018.2869727); 4) a trained convex ridge regulariser (10.1109/TCI.2023.3306100); and 5) the alternative IRL strategy proposed in arXiv:2304.10250. The relative advantages and drawbacks of these approaches relative to the proposed approach should be discussed in detail.**
> >
> > Thank you for the thoughtful suggestions. We would first like to clarify that the primary focus of our work is on deep-learning-based denoisers, which represent a distinct category from traditional denoisers such as BM3D and GMM-based methods. Our goal is to push the area of Plug-and-Play (PnP) approaches by focusing on single-shot learning within this deep-learning framework.
> >
> > That being said, we have taken your suggestions into account and incorporated comparisons with BM3D, which is one of the most widely recognised traditional denoisers. Additionally, we have included comparisons with PnP-DIP (Deep Image Prior), which better aligns with our deep learning-based approach than other traditional methods.
> >
> > Regarding the alternative strategy proposed in arXiv:2304.10250, we would like to clarify that while this paper focuses on a different loss function for optimisation, it uses SIREN as the implicit neural representation (INR). Since we have already included comparisons with SIREN in our ablation study (refer to 4.3), this covers a key aspect of that approach. We believe that the main innovation in INR-based methods, including SIREN, lies in the choice of activation functions—a concept we also explore and propose in our work. This is different in philosofy than that arXiv:2304.10250.
> >
> > However, we provide a more fitting and comprehensive comparison, we have added results for another recent INR-based approach, WIRE, which is among the latest methods in this space. This ensures a balanced evaluation of our method against both traditional and more relevant deep learning-based alternatives.
> >
> > All these changes can be seen in blue colour in the updated manuscript.
> >
> >
> > ➡️ **The statements following (6) regarding the convergence of the scheme should be significantly revised. The current submission does not establish the convergence of Algorithm 1. Instead, I would recommend mentioning the analysis of the convergence of Algorithm 1 as an important perspective for future work. Moreover, I also recommend revising the introduction to mention stochastic plug-and-play Bayesian imaging methods (see for example https://doi.org/10.1137/21M140634) and providing some pointers to this literature.**
> >
> > Thank you for the valuable comment. We would like to clarify that the primary focus of our work is on the convergence of the proposed implicit neural prior. Indeed, the convergence of the broader PnP framework, particularly Algorithm 1, is a significant and intricate topic that goes beyond the scope of this submission and would require a dedicated study. We acknowledge the importance of this aspect and have added it as a direction for future work in the conclusion.
> >
> > Additionally, we appreciate the suggestion regarding stochastic Plug-and-Play Bayesian imaging methods. We have included the relevant references in the introduction and related work sections, specifically incorporating the suggested citation as a complementary one.

---

> > > ### Comment · Reviewer_foFv · 2024-10-05
> > > **Response to Official Response 2**
> > >
> > > I thank the authors for their response and for incorporating a comparison with BM3D.
> > >
> > > Regarding equation (9) in the revised manuscript and the "Convergence of the prior": while I agree with the mathematical statement (9) and with the proof (10), I still believe that the claims derived from this mathematical statement should be curtailed. The convergence properties of optimisation algorithms dealing with (6) will depend on a non-trivial interplay between the mathematical properties of (6) and the characteristics of the algorithm used, which goes well beyond the properties established in (9).

---

> > > > ### Author Response · Authors · 2024-10-07
> > > > **Official Response 3**
> > > >
> > > > Thank you for the comment. To address this, we have explicitly clarified in the manuscript that the convergence results we present are specifically in terms of the behaviour of the prior within our proposed framework. As the reviewer correctly pointed out, a full analysis of the convergence properties of the entire PnP method, considering both the problem formulation and the optimisation algorithm, would require a more comprehensive treatment, which is beyond the scope of this paper and would result in  an entirely separate study.

---

> > ### Comment · Reviewer_foFv · 2024-10-05
> > **Response to Official Response 1**
> >
> > I thank the authors for their response. The revised introduction presents the contributions of this manuscript within the existing plug-and-play literature more clearly, and references the literature more completely. While I perfectly understand that the focus of this paper is on denoisers based on deep-learning and on plug-and-play methods for "single-short" situations, as I explained in my previous review, I do believe that "single-shot" denoising has been already widely studied before deep-learning became the predominant approach for denoising within plug-and-play algorithms, albeit under different names and without using neural networks. Before deep learning became the predominant approach for denoising within plug-and-play algorithms, most SOTA PnP-methods operated in a single-shot manner, but used patch-based representations. These approaches were progressively replaced by deep learning approaches that were better able to leverage large amounts of training data. The results of this manuscript suggest that, at least for some problems, deep learning approaches are also relevant for PnP in situations with no training data available through the use of implicit neural representations. In my opinion, this aspect of the introduction could be explained more clearly.

---

> > > ### Author Response · Authors · 2024-10-07
> > > **Official Response 3**
> > >
> > > Thank you for your  comment. We agree with the reviewer that, prior to the advent of deep learning, plug-and-play (PnP) methods and other frameworks for inverse problems predominantly relied on single-image techniques. Historically, there was a clear distinction between single-image and multi-image approaches. With the introduction of deep learning, the concept of "single-shot" learning  is associated with deep neural network-based techniques. We will clarify this in the introduction to better articulate this distinction.

---

### Decision · Action_Editor_W51q · 2024-10-15

**Recommendation:** Accept as is

**Comment:**

The development of new and efficient denoisers that require limited training data is of significant interest in the field of inverse problems. As such, this paper aligns well with the scope of TMLR.

**Audience:**

The design of new and efficient denoisers which requires a limited training data is of first interest currently for inverse problems. Therefore, the paper fits TMLR scope.

**Claims And Evidence:**

The paper proposes a new single-shot plug-and-play method for solving inverse problems, with a particular focus on image restoration with limited training data. The method consists of two phases: first, it involves learning a denoiser by leveraging implicit neural representations (INR) and a new activation function. By using INR denoisers, the approach requires significantly less data compared to traditional methods like DnCNN or U-Net. In the second phase, this denoiser is integrated into the PNP-ADMM framework. The method shows strong experimental performance in tasks such as image super-resolution and deblurring.